# Observation of giant room-temperature anisotropic magnetoresistance in the topological insulator $\beta$-Ag$_2$Te

Wei Ai[1,4], Fuyang Chen[2,4], Zhaochao Liu[1], Xixi Yuan[3], Lei Zhang[1], Yuyu He[1], Xinyue Dong[1], Huixia Fu[3] ✉, Feng Luo [1] ✉, Mingxun Deng [2] ✉, Ruiqiang Wang[2] & Jinxiong Wu [1] ✉

Achieving room-temperature high anisotropic magnetoresistance ratios is highly desirable for magnetic sensors with scaled supply voltages and high sensitivities. However, the ratios in heterojunction-free thin films are currently limited to only a few percent at room temperature. Here, we observe a high anisotropic magnetoresistance ratio of −39% and a giant planar Hall effect (520 μΩ·cm) at room temperature under 9 T in $\beta$-Ag$_2$Te crystals grown by chemical vapor deposition. We propose a theoretical model of anisotropic scattering − induced by a Dirac cone tilt and modulated by intrinsic properties of effective mass and sound velocity − as a possible origin. Moreover, small-size angle sensors with a Wheatstone bridge configuration were fabricated using the synthesized $\beta$-Ag$_2$Te crystals. The sensors exhibited high output response (240 mV/V), high angle sensitivity (4.2 mV/V/°) and small angle error (<1°). Our work translates the developments in topological insulators to a broader impact on practical applications such as high-field magnetic and angle sensors.

The in-plane anisotropic magnetoresistance (AMR) and planar Hall effect (PHE) are the change of longitudinal ($R_{xx}$) and transverse resistances ($R_{xy}$) of a material, respectively, depending on the angle between the current and applied in-plane magnetic field[1–6], which have been widely used in our daily life, such as magnetoresistive read head, magnetic sensors and memories. The PHE is closely related and shares the same origin with AMR, which manifests itself when the magnetic field, applied current, and the induced transverse Hall voltage all lie in the same plane, precisely in a configuration when the conventional Hall effect disappears. The AMR and PHE were first observed in the ferromagnetic system[7–9] originating from the spin related anisotropic scattering[4]. Recently, a series of topological materials, including Weyl/Dirac semimetals[10–16] and topological insulators (TI)[17–21], were found to

affirmatively show PHE and AMR due to the chiral anomaly[10] and anisotropic backscattering arisen from the tilt of the Dirac cone[22]. Even though PHE and AMR can be observed in a large amount of material systems, it is still highly desirable to achieve large room-temperature AMR ratios in devices for the advantages of fabricating magnetic sensors and memories with scaled operating voltages, small device size and high sensitivity, since the intrinsic AMR ratios from single thin films are always limited to a few percent at room temperature. For example, the saturated AMR ratios of ferromagnetic metals are usually less than 1% or at most up to a few percent (2–3%)[2,7]. Comparatively, the AMR ratios of Weyl/Dirac semimetals, such as Cd$_3$As$_2$ and ZrTe$_{5-\delta}$, were found to be as high as tens of percentage (−68%) at low temperature and high magnetic fields[11,12], but decrease rapidly to zero or a few

[1]Tianjin Key Lab for Rare Earth Materials and Applications, Center for Rare Earth and Inorganic Functional Materials, Smart Sensor Interdisciplinary Science Center, School of Materials Science and Engineering, Nankai University, Tianjin 300350, China. [2]Guangdong Provincial Key Laboratory of Quantum Engineering and Quantum Materials, School of Physics, South China Normal University, Guangzhou 510006, China. [3]Center of Quantum Materials and Devices & College of Physics, Chongqing University, Chongqing 401331, China. [4]These authors contributed equally: Wei Ai, Fuyang Chen. ✉e-mail: hxfu@cqu.edu.cn; feng.luo@nankai.edu.cn; dengmingxun@scnu.edu.cn; jxwu@nankai.edu.cn

percent while warming up to room temperature. Regarding the topological insulators, experimental observations of AMR and PHE were only reliably and simultaneously achieved at low temperature up to now[18–20]. In summary, the experimental realization of a giant AMR and PHE at room temperature is still a big challenge in a single material.

Integration of alternating ferromagnetic and nonmagnetic multilayers with well-defined thicknesses has been verified as an efficient way to greatly improve the AMR ratios of the angle sensors—giant magnetoresistance (GMR) sensors[23–26]. Unlike the ordinary AMR sensors based on single ferromagnetic layer of permalloy, multilayer GMR angle sensors can exhibit large voltage outputs and high sensitivities even at scaled supply voltages. However, owing to the finite value of pinning field of the ferromagnetic layer[27], the GMR angle sensors can only work in a very narrow range of magnetic fields (typically <100 Gauss), and will becomes inapplicable for in-plane angle detection in the presence of a strong magnetic field. Besides, complex heterostructure of ferromagnetic/nonmagnetic multilayers in GMR sensors will undoubtedly add the difficulties for device fabrication and increase the cost. Therefore, magnetic angle sensors with the characteristics of simple structure, high output voltage and wide operating magnetic field range are yet to be developed.

Here, based on the $\beta$-Ag$_2$Te nanoplates grown by chemical vapor deposition (CVD), we achieved the experimental observation of a high AMR ratio up to −39% and giant PHE amplitude (520 $\mu\Omega\cdot$cm) at room

temperature and 9 T. Combined with the theoretical analysis, the origin of observed giant room-temperature PHE and AMR may be attributed to the intrinsic properties of $\beta$-Ag$_2$Te, such as low sound velocities and small effective masses. Furthermore, a high-field AMR angle sensor with a small device size was successfully fabricated by etching a single nanoplate of $\beta$-Ag$_2$Te into the Wheatstone bridge structure. More importantly, the $\beta$-Ag$_2$Te AMR angle sensors simultaneously own simple device structure, high output response (240 mV/V, almost one order of magnitude higher than commercial AMR sensors), high angle sensitivity (4.2 mV/V/°), small angle error (<1°) and wide operating magnetic field range.

## Results

### Structure, CVD growth and characterization of $\beta$-Ag$_2$Te

Monoclinic $\beta$-Ag$_2$Te is a thermodynamically stable phase of Ag$_2$Te, whose crystal structure can be regarded as periodic stacking of triple atom columns in an Ag-Ag-Te or Te-Ag-Ag sequence along $a$ direction (Fig. 1a). To investigate the band structure and phonon dispersion of $\beta$-Ag$_2$Te, we performed density functional theory (DFT) calculations (Fig. 1b–d). $\beta$-Ag$_2$Te possess a small bulk band gap of ~35 meV, and its nontrivial topological nature with a nontrivial invariant $Z_2 = 1$ is confirmed by analyzing the parity of occupied Bloch wave functions at the time-reversal invariant points in Brillouin zone[28], which undoubtedly support gapless Dirac-type surface states inside the band gap (Fig. 1b).

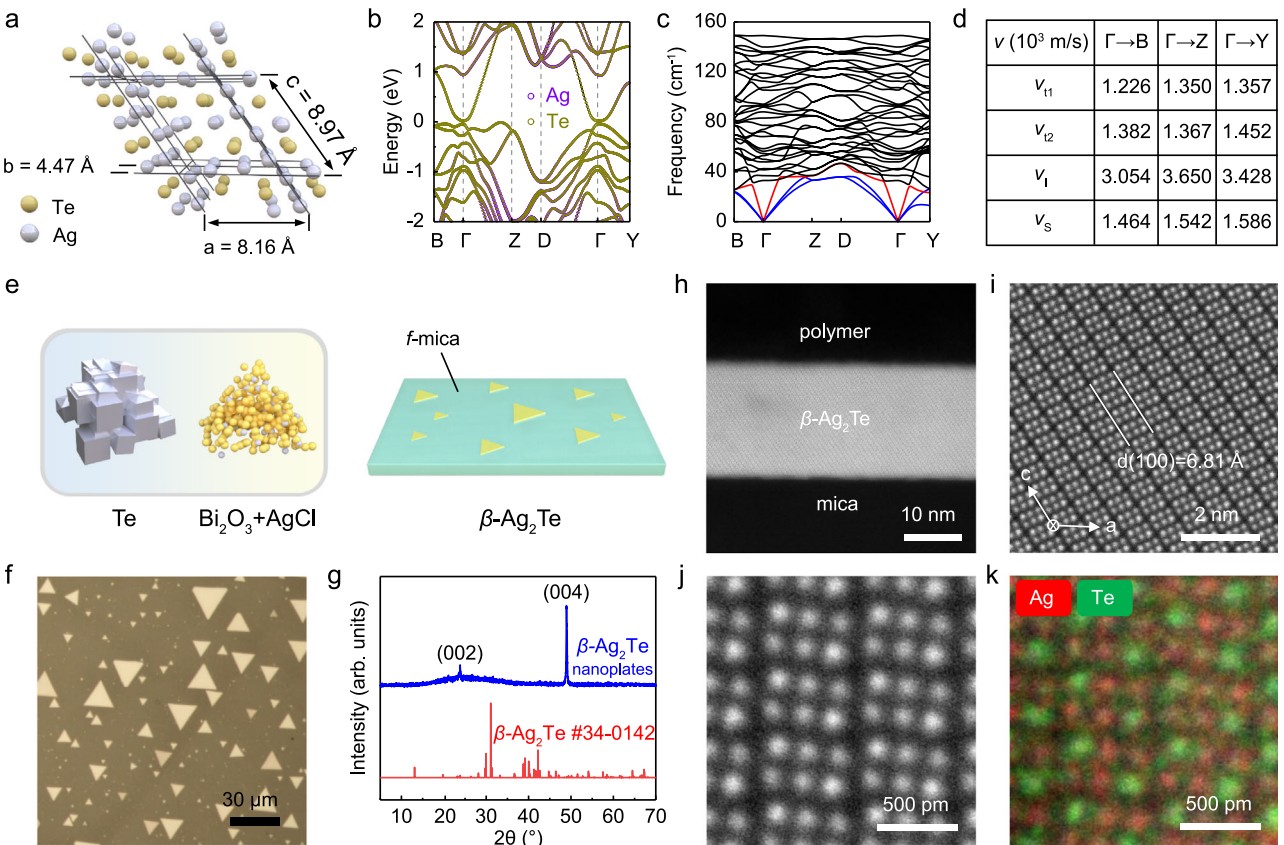

**Fig. 1 | Structure, growth and characterization of $\beta$-Ag$_2$Te topological insulator. a** Crystal structure of $\beta$-Ag$_2$Te with periodic arrangement of 3 atom columns along $a$ direction (P21/C, $a = 8.162$ Å, $b = 4.467$ Å, $c = 8.973$ Å, $\beta = 124.15°$). **b** The calculated band structure of $\beta$-Ag$_2$Te with a small band gap of ~35 meV. **c** The calculated phonon dispersion relation for $\beta$-Ag$_2$Te, in which the dispersions of transverse ($v_{t1}$, $v_{t2}$) and longitudinal ($v_l$) acoustic phonons in the low frequency region were labelled by blue and red lines, respectively. **d** The calculated sound velocities along Γ-B, Γ-Y and Γ-Z directions. **e** Illustration of the chemical vapor deposition (CVD) growth process to synthesize $\beta$-Ag$_2$Te crystals on mica substrate

by using Te lumps and mixed powders of AgCl and Bi$_2$O$_3$ as evaporation sources. **f** Typical optical microscopy (OM) image of as-synthesized $\beta$-Ag$_2$Te crystals on mica. **g** X-ray diffraction (XRD) pattern of as-grown $\beta$-Ag$_2$Te crystals transferred onto the glass substrate. **h, i** Cross-sectional low-magnification (**h**) and high-magnification (**i**) high angle annular dark-field (HAADF) images of a $\beta$-Ag$_2$Te nanoplate on mica prepared by focused ion beam technique. **j, k** The atomic-resolved HAADF image (**j**) and corresponding elemental mapping (**k**) images, both of which clearly distinguished the atomic positions of Ag and Te columns in $\beta$-Ag$_2$Te.

Moreover, the phonon dispersion calculations on $\beta$-Ag$_2$Te were demonstrated in Fig. 1c, from which theoretical sound velocities of 1226, 1382 and 3054 m s$^{-1}$ were extracted along the Γ-B direction for two transverse ($v_{t1}$, $v_{t2}$) and one longitudinal ($v_l$) acoustic phonon, respectively. Notably, the calculated averaged sound velocities ($v_s$) for all the three directions (Γ-B, Γ-Y and Γ-Z) are always around 1500 m s$^{-1}$, which is one of the lowest values among the current material system[29]. This value also matches well with the experimental value and accounts for the ultralow lattice thermal conductivity (~0.4 W m$^{-1}$ K$^{-1}$) observed in $\beta$-Ag$_2$Te[30-32].

Compared to rough polycrystalline films, single-crystalline thin films with ultrasmooth surface are preferable for fundamental research and electronic applications. Up to now, most of works mainly focus on the polycrystalline $\beta$-Ag$_2$Te films[33,34]. Very few were reported on the synthesis and related basic properties of the single-crystalline thin films[35-38]. Here, we developed a facile chemical vapor deposition (CVD) method to synthesize the single-crystalline thin films of $\beta$-Ag$_2$Te. As illustrated in Fig. 1e and Supplementary Fig. 1, the Te lumps and mixture of AgCl and Bi$_2$O$_3$ powders, separately located in two different heating zones, were chosen as evaporation sources. Interestingly, we found Bi$_2$O$_3$ played a key role in the synthesis of $\beta$-Ag$_2$Te, since merely no nanoplates were obtained on mica substrate if no Bi$_2$O$_3$ powders were used under the same CVD condition (Supplementary Fig. 2). Typically, the CVD-grown $\beta$-Ag$_2$Te nanoplates with a thickness of several tens of nanometers exhibited triangular-like or hexagonal-like morphologies, whose domain sizes can be as large as several tens of microns (Fig. 1f and Supplementary Figs. 3, 4). Furthermore, the crystalline phase and valence states of CVD-grown $\beta$-Ag$_2$Te nanoplates were verified by X-ray diffraction (XRD, Fig. 1g), ab-plane transmission electron microscopy (TEM) imaging (Supplementary Fig. 5) and X-ray photoelectron spectroscopy (XPS, Supplementary Fig. 6), all of which matched well with the monoclinic phase. To identify the fine crystal structure of $\beta$-Ag$_2$Te in detail, cross-sectional spherical aberration corrected scanning TEM (STEM) measurements were performed. As depicted in Fig. 1h, i, the signature of periodic stacking of triple atom layers along [100] direction fits well with the ac plane of $\beta$-Ag$_2$Te. Besides, the interface between the epilayer and substrate has no buffer layer at all, thus resulting in nearly perfect atomic ratio of ~2: 1 determined by energy dispersive X-ray spectroscopy (EDX, Supplementary Fig. 7 and Table 1). One step further, we used the technique of atomic-resolved EDX to see the exact locations of Te and Ag atoms in the CVD-grown thin films, which undoubtedly showed the ordered sequence of Ag-Ag-Te or Te-Ag-Ag in Fig. 1j, k. It should be noted that small stoichiometric deviation induced by trace amount of defects, such as 0.01%−0.001% (roughly estimated by the Hall carrier density, Supplementary Fig. 8), may also exist in the CVD-grown samples considering the detection limit and accuracy of EDX. To investigate the potential microscopic conduction mechanism in $\beta$-Ag$_2$Te, we performed DFT calculations to see the formation energies and dopant type for four possible different defects (for details, see Supplementary Fig. 9).

### Giant room-temperature PHE and AMR in CVD-grown $\beta$-Ag$_2$Te

The successful synthesis of $\beta$-Ag$_2$Te single-crystalline thin films greatly facilitates the investigation of its basic properties such as the PHE and in-plane AMR, which have never been reported in $\beta$-Ag$_2$Te before. Here, the standard six-terminal Hall bar configuration was adopted to measure the longitudinal resistance ($R_{xx}$) and Hall resistance ($R_{xy}$) simultaneously. Figure 2a shows the schematic illustration of the setup for PHE and AMR measurements, where the magnetic field ($B$) is applied along the $y$ axis and the current or device rotates in the $x$-$y$ plane. In general, two potential misalignments are present in such a setup: (1) the magnetic field may not strictly locate in the exact sample plane; (2) the Hall voltage legs are not perfectly aligned during the fabrication[9,12]. After averaging the $R_{xy}$ measured under positive and negative magnetic fields and subtracting a constant

offset, the intrinsic PHE signal can be obtained (Supplementary Fig. 10). Prior to the measurement of PHE and in-plane AMR, we conducted ordinary Hall measurements on a 79.5-nm-thick CVD-grown $\beta$-Ag$_2$Te with magnetic field applied perpendicularly to the device plane, showing a very high Hall mobility of ~8000 cm$^2$ V$^{-1}$ s$^{-1}$ and low carrier density of $1.0 \times 10^{18}$ cm$^{-3}$ at room temperature (Supplementary Fig. S8). Subsequently, the magnetic field was applied parallel to this specific device to conduct the planar Hall and in-plane AMR measurements. Figure 2b shows the symmetrized $R_{xy}$ versus $\varphi$ measured at 300 K and 9 T, where the $\varphi$ is the angle between the directions of in-plane magnetic field and current. The PHE follows a sine function with a period of 180° and can be well fitted by the equation $R_{xy} = \gamma \frac{R_{\parallel} - R_{\perp}}{2} \sin 2\varphi$, where $R_{\parallel}$ and $R_{\perp}$ are the $R_{xx}$ when $\varphi$ is equal to 0° and 90°, respectively, and $\gamma$ is geometric ratio of the width to the length of the Hall bar device. In addition, we also measured the angular dependence of PHE over a wide $B$ range at 300 K. Figure 2c shows the PHE amplitude exhibits a monotonically increasing behavior up to ~280 μΩ·cm with increasing field, and no sign of saturation is observed with $B$ up to 9 T. Notably, the room-temperature PHE amplitude in another Hall bar device can even be as large as ~520 μΩ·cm at 9 T (Supplementary Figs. 11, 12), which is much higher than ever reported values in conventional ferromagnetic metals (<1 μΩ·cm) and Dirac semimetals (<40 μΩ·cm, See Supplementary Table 2). In other words, unexpected giant room-temperature PHE in TI was reliably observed in CVD-grown $\beta$-Ag$_2$Te. Besides, the PHE amplitude showed an abnormal temperature dependence, exhibiting unconventional increase in PHE amplitude with temperature increasing over a wide temperature range (Fig. 2d). This phenomenon is totally different from other PHE materials systems induced by ferromagnetism and chiral anomaly[9,12], whose PHE amplitude decreases monotonically upon warming up to room temperature.

Giant room-temperature PHE amplitude guarantees the observation of giant room-temperature AMR ratio in $\beta$-Ag$_2$Te, which is one of the most important factors for the applications in magnetic sensors. Here, the AMR ratio is defined as $\frac{R_{xx,\varphi} - R_{xx,\perp}}{R_{xx,\perp}} \times 100\%$. Because the $R_{xx,\perp}$ is larger than the $R_{xx,\parallel}$, the AMR ratio is negative for $\beta$-Ag$_2$Te. Figure 2f presents the room-temperature angular dependence of AMR ratios at various magnetic fields. It is worth noting that the AMR ratio fits well with the theoretical $\cos^2\varphi$ dependence by reaching its maximum and minimum at 0° and 90°, signifying the ignorable additional contributions caused by misalignments at room temperature in this device. With increasing magnetic fields, the room-temperature AMR ratio increases monotonically and reaches a giant value of −27% at 9 T without the sign of saturation (Fig. 2g). Higher AMR ratio is expected to be observed at higher fields. Moreover, the temperature dependence of AMR amplitude (defined as $R_{xx,\perp} - R_{xx,\parallel}$) was also investigated, showing a similar unconventional increase upon warming up (Supplementary Fig. 13). To check whether the giant room-temperature AMR ratios widely exist in $\beta$-Ag$_2$Te, we performed AMR measurements on 7 different samples from different CVD growth batches at room temperature. As shown in Fig. 2f, all of them exhibited a giant AMR ratio ranging from −24% to −39%. It is worth noting the AMR ratio of −39% is high relative to previously reported values, including the well-known commercial permalloy[39] (~2%), La-Sr-Mn-O[40] (~1%), Dirac/Weyl semimetals[12] (up to ~4% at 9 T). Remarkably, the giant AMR and PHE behavior in CVD-grown $\beta$-Ag$_2$Te is also very robust to air exposure. As shown in Supplementary Fig. 12, very small change of AMR ratio and PHE were obtained on the same device before and after being exposed to air for 1 month.

### Possible origin of giant room-temperature PHE and AMR in $\beta$-Ag$_2$Te

To investigate the possible origin of giant room-temperature PHE and AMR in $\beta$-Ag$_2$Te, we perform the following theoretical calculation. We

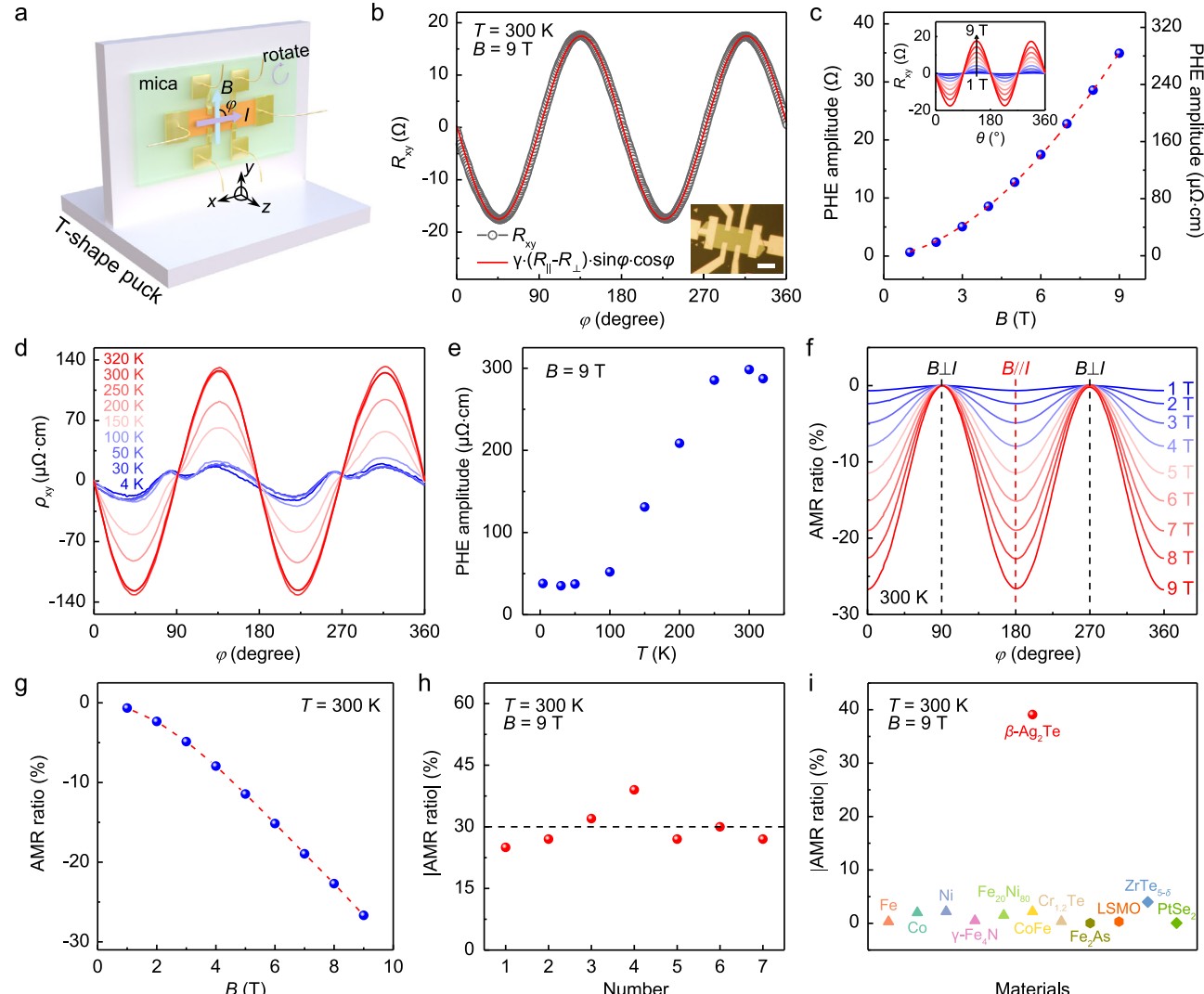

**Fig. 2 | Giant room-temperature planar Hall effect (PHE) and anisotropic magnetoresistance (AMR) in β-Ag₂Te nanoplate. a** Schematic demonstration for measuring the AMR and PHE of CVD-grown β-Ag₂Te, in which the Hall-bar device was placed on a *T*-shape puck with a rotator. **b** Angular dependence of planar Hall resistance ($R_{xy}$) measured at 300 K and 9 T, showing a periodicity of π and a nice fit to the inset equation. The inset is the OM image of the fabricated Hall-bar device, and the scale bar is 10 μm. **c** Magnetic-field dependence of the PHE amplitude ($\Delta\rho_{xy}$, defined as $\rho_{xy}$ ($\varphi = 135°$) $-\rho_{xy}$ ($\varphi = 45°$)) measured at 300 K; the inset shows the raw $R_{xy}$ -$\varphi$ data measured at various in-plane magnetic fields ranging from 1 to 9 T. The red dashed line is a visual guide. **d** Temperature-dependent planar Hall

measurements ($R_{xy}$-$\varphi$) from 320 to 4 K. The in-plane magnetic field is kept constant as 9 T. **e** The extracted PHE amplitude as a function of temperature on the basis of (**d**). **f** Angular dependence of AMR ratio measured in different magnetic fields and at 300 K. **g** The extracted magnetic-field dependence of AMR ratio at 300 K, which can reach a high value of −~27% at 9 T. The red dashed line is a visual guide. **h** The statistic of AMR ratios from 7 different β-Ag₂Te Hall-bar devices, showing a giant room-temperature AMR ratio of −~39%. The black dashed line is a visual guide. **i** Comparison of the AMR ratio in β-Ag₂Te with the reported values in other typical ferromagnetic metals, topological semimetals and topological insulators.

consider energies within the TI bulk band gap where only the surface states are relevant and the Hamiltonian[22] reads

$$H = \sum_{K} c_k^\dagger H_k c_k + \sum_{q} \hbar\omega_q b_q^\dagger b_q + \sum_{kq} D_q \left( b_q + b_{-q}^\dagger \right) c_{k+q}^\dagger c_k + \sum_{kq} c_{k+q}^\dagger U_q c_k \quad (1)$$

Here the first term describes the Dirac fermions of the TI surface and the second term captures the motion of the phonons due to lattice vibration, in which $c_k(b_q)$ represents the electron (phonon) annihilation operator and $\omega_q = \upsilon_s q$ is the phonon dispersion with $\upsilon_s$ being the velocity of sound. The last two terms characterize the electron-phonon and electron-impurity scattering, respectively. In the presence of magnetic field **B**, the single-particle excitation of the TI surface can be described by the low-energy effective Hamiltonian $H_k = \frac{\hbar^2 k^2}{2m_{eff}} + \hbar\upsilon_F(k_x\sigma_y - k_y\sigma_x) - g\mu_B\mathbf{B}\cdot\boldsymbol{\sigma}$ with g and $\mu_B$ being the Landé

g-factor and Bohr magneton, where $m_{eff}$ is the effective mass of the band electrons induced by the particle-hole asymmetry, $\boldsymbol{\sigma} = (\sigma_x, \sigma_y, \sigma_z)$ denotes the spin Pauli matrix and $\upsilon_F$ is the Fermi velocity.

For simplicity, it is assumed that the magnetic field is fixed in the x-y plane. By diagonalizing $H_k$, we can obtain the dispersion $\varepsilon_{k,\eta} = \frac{\hbar^2 k^2}{2m_{eff}} + \eta\hbar\upsilon_F |\mathbf{k} - \mathbf{k}_B|$, where $k_B = \frac{g\mu_B}{\hbar\upsilon_F}(B_y, -B_x)$ and $\eta = \pm1$ labels the helicity of the Dirac fermions. As a result of the particle-hole asymmetry, the Dirac cone will be tilted by the in-plane magnetic field when the Dirac point is shifted away from **k** = 0 to **k** = **k**$_B$ as illustrated in Fig. 3a. Accompanied with this, the Fermi surface deforms to an ellipse and the spin texture will change with the magnetic field, which causes the anisotropic electron scattering and thus results in the appearance of PHE in TI. The amplitude of the PHE will show unconventional increase with increasing temperature as the electron scattering is dominated by the electron-phonon interaction. To

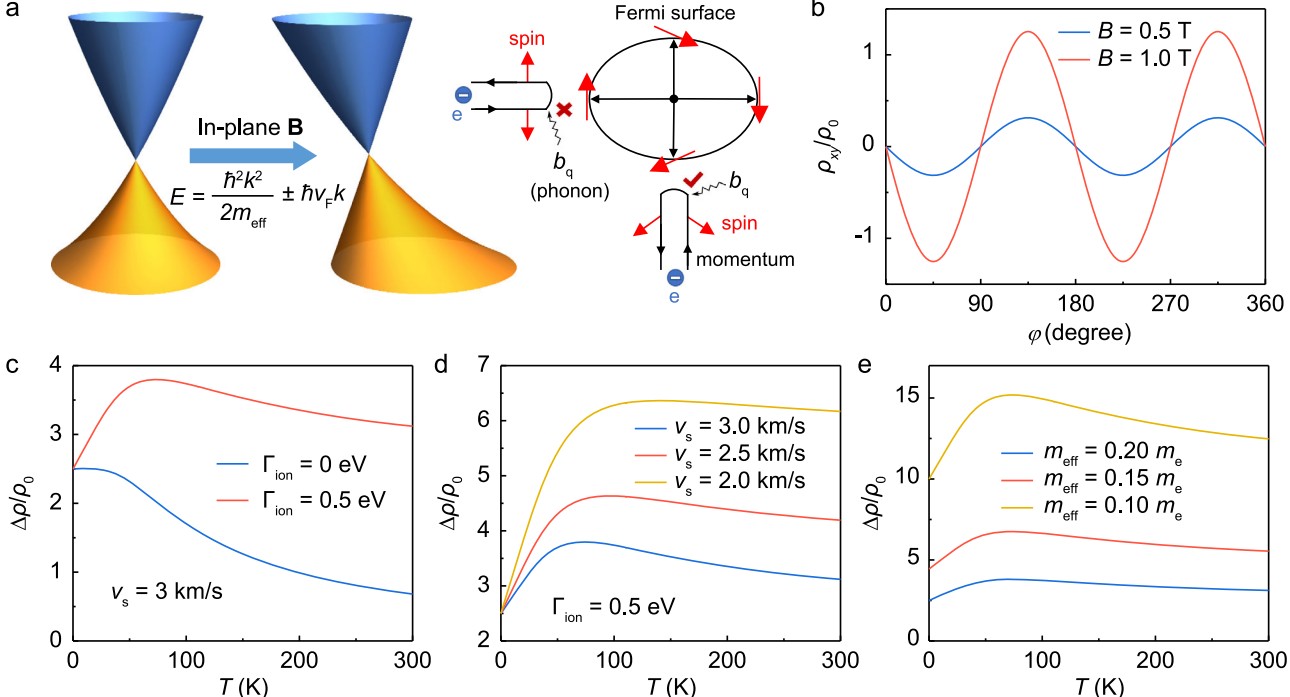

**Fig. 3 | Theoretical understanding of the rule on regulating PHE amplitudes in topological insulators. a** Schematic demonstration for the surface Dirac cone tilted by the in-plane magnetic field, which modifies the spin texture and results in the anisotropic electron scattering on the Fermi surface (closed black ellipse). The $b_q$ associated with the wavy line represents the phonon annihilation process (absorbed by an electron) to cause a forbidden or allowed back-scattering event. The red and black arrows are the spin and momentum directions of the surface states, respectively. **b** Angular dependence of the Hall resistivity. **c–e** The PHE amplitude $\triangle\rho$ vs temperature for varied electron-phonon interaction strength (**c**), sound velocities (**d**) and effective masses (**e**), respectively.

demonstrate this point, we derive the Green's functions for the TI surface states and calculate the conductivity by the Kubo–Středa formula[41–44] (for details, please see the discussion part in supporting information). The resulting Hall resistivity can be expressed as $\rho_{xy} = \triangle\rho \sin\theta_B \cos\theta_B$, where $\theta_B$ is the angle of the magnetic field and

$$\triangle\rho = \frac{1}{F(T)} \frac{m_e^2}{m_{eff}^2} B^2 \rho_0 \tag{2}$$

is the PHE amplitude. For convenience, we here set $\rho_0 = \frac{h}{e^2}\left(\frac{g\mu_B[B=1\,\text{Tesla}]}{m_e v_F^2}\right)^2$ as the unit of resistivity. The temperature factor is $F(T) = -\int_{-\infty}^{\infty} d\varepsilon\,\partial_\varepsilon f(\varepsilon) \frac{|\varepsilon|}{\Gamma(\varepsilon)}$ with $\Gamma(\varepsilon) = \Gamma_0 + \Gamma_{ion} \frac{k_B T}{2Mv_s^2} \frac{\zeta}{\sinh\zeta}$ as the lifetime broadening function due to the electron scattering (Supplementary Fig. 14). Here, $\zeta = \frac{v_s}{v_F} \frac{|\varepsilon|}{k_B T}$, $\Gamma_0 = \frac{\hbar}{\tau_{im}}$ and $\Gamma_{ion} = \frac{\hbar}{\tau_{ion}}$, with $\tau_{im}$ and $\tau_{ion}$ being respectively the averaged electron-impurity and electron-ion interaction time.

As shown in Fig. 3b, the tilt of Dirac cone induced by in-plane magnetic field can indeed cause the PHE with a period of 180°. Moreover, Fig. 3c shows that without the electron-phonon interaction, the PHE amplitude decreases monotonously with temperature. However, the PHE amplitude exhibits a non-monotonic temperature dependence with a peak developing at $T = T_P$ when the electron-phonon scattering is taken into consideration. This unconventional temperature dependence of PHE amplitude originating from the surface states fits well with the one observed in $\beta$-Ag$_2$Te (Fig. 2e) and other TIs[19,20]. More importantly, as elucidated in Fig. 3d, e, the value of $T_P$ and PHE amplitude can be greatly increased by regulating the TI's inherent properties, such as reducing the sound velocities and effective masses (Supplementary Fig. S15). In other words, we theoretically confirmed the possible existence of giant room-temperature PHE and AMR ratio in TIs, especially when they have small effective masses and low sound

velocities, just like the case in $\beta$-Ag$_2$Te. Except for the ultralow sound velocities mentioned above (Fig. 1c), $\beta$-Ag$_2$Te is also a well-known topological insulator for its high mobility[36] and ultralow effective mass[38] (-0.08 $m_0$), which match well with the theoretical demand for achieving the giant room-temperature PHE and AMR in TIs. It should be noted that our model is a qualitative analysis, rather than a quantitative tool, to see how the PHE amplitude changes with the parameters and thus help us to understand and figure out the possible origin of large room-temperature AMR ratio and PHE in $\beta$-Ag$_2$Te system. Our theoretical results are qualitatively consistent with the experimental measurements, which enable the investigation of the underlying physics. However, we can't exclude that other models may also give the similarly consistent results, which need further investigations in the future.

We should emphasize that even though the bulk conduction of topological insulator can also induce a PHE in theory, it will show a conventional temperature dependent PHE amplitude instead, namely having a high PHE amplitude at low temperature but vanishing to near zero at room temperature (Supplementary Fig. S16), which is apparently different from the one observed in Fig. 2e. Therefore, considering the low carrier density and unconventional temperature dependence on PHE amplitude, we believe the surface states of $\beta$-Ag$_2$Te still dominate the charge transport at room temperature, even though a small amount of bulk states will inevitably contribute to the charge conduction.

### High-field AMR magnetic and angle sensor based on $\beta$-Ag$_2$Te
The giant room-temperature AMR ratio and compatibility with high magnetic field in $\beta$-Ag$_2$Te make it highly appealing for the application of low-power magnetic sensors for angle and field detection over a wide magnetic field range. Owing to the superior stability and tolerance to external fields, the widely used Wheatstone bridge configuration[45–49], composed of 4 perpendicularly stacked equivalent

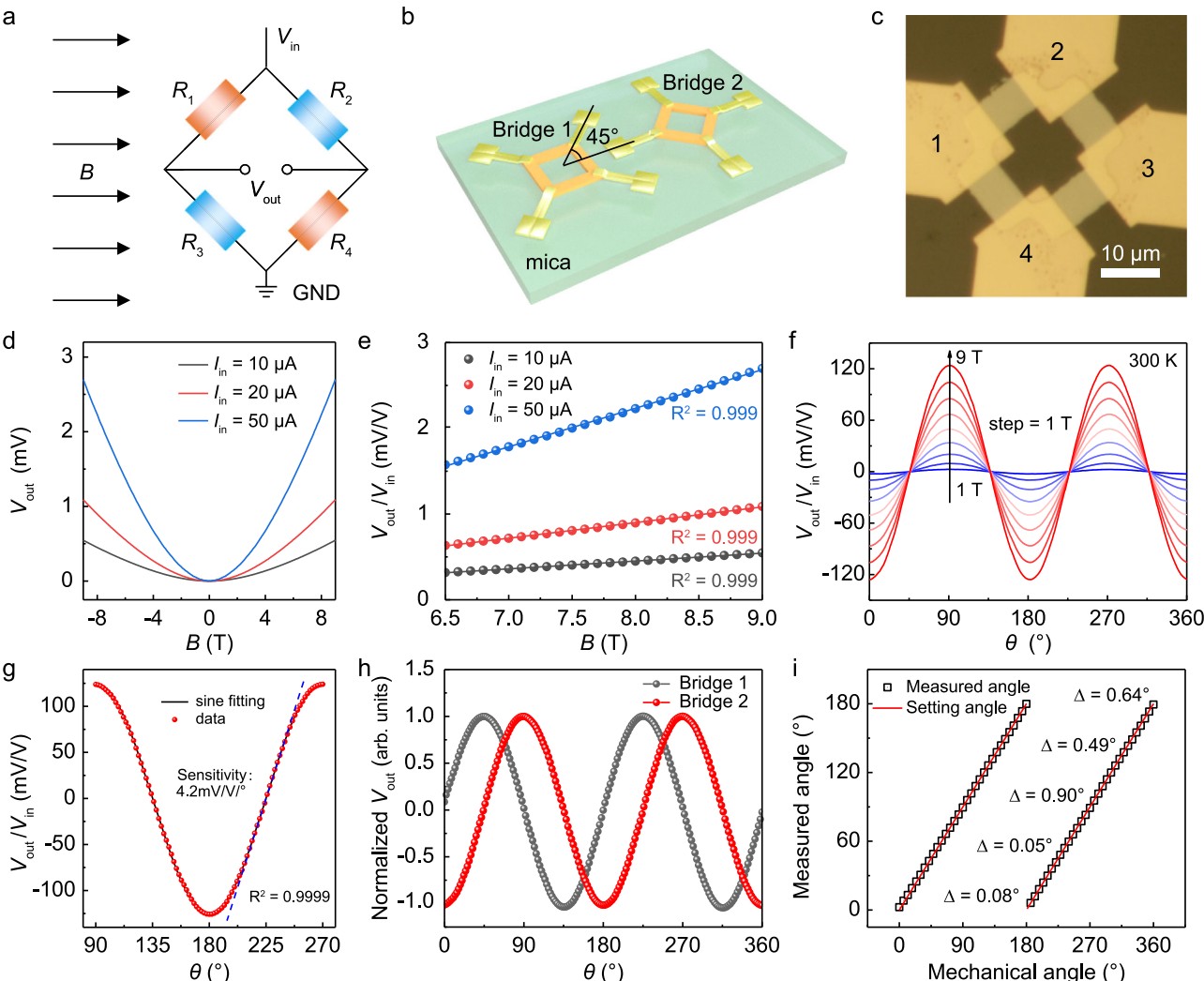

**Fig. 4 | β-Ag₂Te based high-field AMR magnetic sensor with high output voltage and angular accuracy. a** Schematic diagram of the structure of Wheatstone bridge in AMR angle sensor. **b** Illustration of the β-Ag₂Te AMR angle sensor on mica substrate, which consists of two Wheatstone bridges arranged at 45° from each other. **c** Typical OM image of an as-fabricated β-Ag₂Te AMR angle sensor, which was directly patterned into a structure of Wheatstone bridge on mica by wet chemical etching with dilute HNO₃. The probes 1 and 3 are defined as current supply terminals, and probes 2 and 4 are used to measure the output voltage ($V_{out}$). **d, e** Output response of the device as a function of magnetic field under different input currents ($V_{in}$ = 10, 20, 50 μA, **d**), showing linear dependence in high magnetic field region (**e**). The $B$ was parallel to one arm of the Wheatstone bridge. **f** Angle-dependent output-to-input voltage ratio ($V_{out}/V_{in}$) at different magnetic fields ranging from 1 to 9 T (step: 1 T). **g** The sinusoidal curve fitting of an output waveform from 90° to 270° ($B$ = 9 T), giving a goodness of fit of 0.9999 and high angle detection sensitivity of 4.2 mV/V/°. **h** The normalized outputs of two AMR bridges rotated at 45° from each other, showing a 90° phase shift between two sinusoidal outputs. **i** The measured angles versus mechanical angles, showing a perfect linear dependence and small angular error (Δ) of <1° over a 180° measurement range. The measured angles were obtained by performing an arctangent calculation based on two sinusoidal outputs in (**h**).

resistors (Fig. 4a), was adopted to fabricate the AMR sensor of β-Ag₂Te on mica substrate. Besides, by integrating two sensing elements on the same substrate rotated at 45° from one another (Fig. 4b), the AMR sensor can be used over a full 180° measurement range. To eliminate the possible variations of electrical properties from sample to sample, the AMR sensor was successfully fabricated by directly etching one β-Ag₂Te nanoplate into four individual elements assisted by dilute nitric acids (Supplementary Fig. 17). The typical optical image of single Wheatstone bridge on β-Ag₂Te is shown in Fig. 4c. It should be emphasized here that the device size of 15 × 15 μm² is much smaller than the commercial angle sensors (typically > 1 mm²) with complex barber pole structure[45,50], suggesting the potential and feasibility of device miniaturization. Unlike the GMR sensor composed of multilayer ferromagnetic/nonmagnetic films with well-defined thicknesses[49], the device configuration of our AMR sensor is very simple with non-encapsulated β-Ag₂Te as the single functional layer.

In general, the main functions of AMR magnetic sensor are detection of field strength and rotated angles. First, we evaluated its potential as magnetometer by measuring the relation between output voltages ($V_{out}$) and magnetic field strength at room temperature. As elucidated in Fig. 4d, e, the $V_{out}$ showed a quadratic-like dependence at relatively small $B$, but showed a linear behavior when $B$ > 6.5 T, thus can be functionalized into high-field magnetometers. It is worth noting the detection range of the β-Ag₂Te based magnetometers (at least several Tesla) is much wider than the commercial AMR sensor of permalloy, which is usually limited to 0–20 Oe due to the low saturation field[47]. Subsequently, low-power AMR angle sensor with high accuracy was demonstrated based on β-Ag₂Te. As displayed in Fig. 4f, the angle-dependent outputs, defined as the ratio of $V_{out}$ and input voltage $V_{in}$ ($V_{out}/V_{in}$), were measured under different magnetic fields. Remarkably, the peak output can be as large as 240 mV/V at 9 T, suggesting a room-temperature AMR ratio of ~24% even after chemical etching. This value is about one order higher than commercial permalloy angle sensors,

and comparable to typical GMR sensor but can be operated over a much wider magnetic field range. It is worth noting that higher peak outputs mean lower supply voltages and power consumption are needed to obtain the similar accuracy. Besides, the outputs of the sensor showed a perfect sinusoidal fitting with a goodness of 0.9999 and high sensitivity of 4.2 mV/V/° for angle detection (Fig. 4g). In order to achieve a full 180° measurement range, two AMR bridges rotated at 45° from each other were integrated together to achieve two sinusoids with a relative phase shift of 90° (Fig. 4h). Therefore, the absolute measured angle ($\theta_{measured}$) over 180° can be obtained by performing an arctangent calculation based on the following equation

$$\theta_{measured} = \frac{\arctan 2(\frac{V_{sin}}{V_{cos}})}{2} \qquad (3)$$

As a result, nearly perfect linear response between the measured angles and mechanical angles ($\theta_{setting}$) were obtained (Fig. 4i and Supplementary Fig. 18), from which a small angular error ($\Delta$) of less than 1° was extracted by the equation $\Delta = |\theta_{measured} - \theta_{setting}|$. The angular accuracy of our preliminary version of AMR sensor is already comparable to many commercial angle sensors after careful optimizations.

In summary, we experimentally and theoretically confirmed the possible existence of giant AMR and PHE in topological insulator at room temperature. The observed giant room-temperature AMR ratio in CVD-grown $\beta$-Ag$_2$Te greatly facilitates the fabrication of AMR sensors with high output voltage at scaled supply voltage. Combined with the feature of heterojunction-free device structure, feasibility to device miniaturization and wide operating magnetic field range, our work demonstrates that the $\beta$-Ag$_2$Te AMR sensor may act as an attractive complement to current commercial AMR and GMR sensors, and show apparent advantages for angle and magnetic field detections at strong magnetic fields.

## Methods

### CVD growth and characterization of $\beta$-Ag$_2$Te nanoplates
2D $\beta$-Ag$_2$Te crystals were synthesized in a homemade low-pressure CVD system with dual-heating zones. Bulk of Te (Alfa Aesar) and the mixture of Bi$_2$O$_3$ (Macklin) and AgCl (Damas) were placed in the upstream and downstream heating zones, and the mica substrates were located close to the downstream heating center (1–2 cm away). Typically, the evaporated temperatures of Te and Bi$_2$O$_3$ + AgCl were kept at 420 °C and 640 °C, and 200 sccm Ar was employed as the carrier gas. The pressure was kept constant at 400 Torr during the whole CVD growth process.

To characterize the as-grown $\beta$-Ag$_2$Te nanoplates, optical microscopy (OM, WY-910), atomic force microscopy (AFM, Bruker Dimension Icon), transmission electron microscopy (TEM, JEM 2800), and aberration-corrected scanning transmission electron microscopy (AC-STEM, FEI-Titan Cubed Themis G2 300 operating at 300 kV) were used. The cross-sectional TEM sample was prepared by focused ion beam (FEI, Helios 5 CX).

### DFT calculation for $\beta$-Ag$_2$Te
First-principles calculations were performed using the Vienna ab initio simulation package (VASP) based on density functional theory (DFT). The generalized gradient approximation (GGA) of Perdew-Burke-Ernzerhof (PBE) was adopted for the exchange-correlation functional. The projector-augmented-wave (PAW) pseudopotential was implemented with an energy cutoff of 400 eV as the basis set. A monoclinic crystal structure with the experimental lattice constants was adopted. In structural optimizations, all the atoms were fully relaxed until the residual force on each relaxed atom was less than 0.01 eV/Å within the energy convergence threshold of $10^{-5}$ eV. A $\Gamma$-centered Monkhorst−Pack $k$-point mesh of $7 \times 11 \times 7$ was adopted for sampling

the first Brillouin zone. The band structure and topological invariance $Z_2$ were obtained with spin-orbit coupling (SOC) considered. The phonon spectrum was carried out by the density-functional perturbation theory (DFPT) method via the Phonopy package. Averaged velocities ($v_s$) were obtained according to the equation, $v_s = (\frac{1}{3}[\frac{1}{v_{t1}^3} + \frac{1}{v_{t2}^3} + \frac{1}{v_l^3}])^{-\frac{1}{3}}$, where the $v_{t1}$, $v_{t2}$ represent the two transverse sound velocities and $v_l$ is the calculated longitudinal sound velocity.

### Region-selective wet-chemical etching of $\beta$-Ag$_2$Te
The as-synthesized $\beta$-Ag$_2$Te nanosheets on mica substrate can be patterned into predesigned geometry by the region-selective chemical etching process with the assistance of photolithography. First, the location markers were pre-patterned using the standard UV photolithography process. Next, a layer of photoresist (AR-P-5350) was spin-coated onto the surface of $\beta$-Ag$_2$Te, followed by a patterning process using the maskless laser direct writing system (Microlab-III). After developing, the oxygen plasma treatment (30 W, 5 s) was employed to remove the possible polymer residual on top surface of $\beta$-Ag$_2$Te. Last, the sample was immersed in HNO$_3$ solution (7.6 mol/L) for about one minute, then transferred to deionized water immediately to terminate the etching process. With the process above, the $\beta$-Ag$_2$Te nanosheet with the targeted Wheatstone bridge configuration can be obtained after removing the AR-P-5350 layer with hot acetone.

### Device fabrication and electrical transport measurements
The six-terminal Hall-bar devices used to measure the PHE and AMR in Fig. 2 were fabricated by the following process. First, the location markers were pre-patterned on mica substrate using the standard UV photolithography process. Then, electron-beam lithography (EBL) was used to write six electrode legs of the Hall bar device on CVD-grown $\beta$-Ag$_2$Te, followed by thermal evaporation of the contact metals Pd/Au (5/70 nm). The device fabrication process to fabricate the $\beta$-Ag$_2$Te based AMR sensor is the same with the Hall-bar device, rather than 6-electrode legs but only 4 electrode legs are written instead.

The PHE and AMR data in Fig. 2 and AMR sensing data in Fig. 4 were collected in the Physical Properties Measurement Systems (PPMS-9T, Quantum Design) coupled with a home-made electrical measurement system, composed of two Keithley 2400 meters, two Keithley 2182 A nanovoltmeters and a Keithley 6221 AC and DC current source. The Hall-bar and Wheatstone bridge devices on mica substrate were loaded onto a $T$-shape puck to make the magnetic field rotate in the substrate plane with an angle range of −10–370°.

## Data availability
Relevant data supporting the key findings of this study are available within the article and the Supplementary Information file. All raw data generated during the current study are available from the corresponding authors upon request.

## Code availability
The codes used in the data analysis are available from the corresponding authors upon request.

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

## Acknowledgements

We thank Prof. Hongtao Yuan @Nanjing University, Dr. Wenjie Zhang and Dr. Jingyue Wang @Peking University for fruitful discussion. The work was supported by the National Natural Science Foundation of China (No. 92064005, No. 12274146, No. 12174121, No. 12104072), the National key research and development program of China (No. 2021YFA1601004),

the PhD Candidate Research Innovation Fund of NKU School of Materials Science and Engineering, the Opening Project of State Key Laboratory of High Performance Ceramics and Superfine Microstructure (SKL202211SIC), the Guangdong Basic and Applied Basic 297 Research Foundation under Grant No. 2023B1515020050.

## Author contributions

J. Wu convinced the original ideals and supervised the whole project. W. Ai performed the CVD growth, characterization, device fabrication and electrical measurements under the assistance of Z. Liu, L. Zhang, Y. He and X. Dong. The theoretical calculation was performed by F. Chen under the supervision of M. Deng and R. Wang, and Z. Liu plotted the cartoon diagram. X. Yuan performed the DFT calculation under the supervision of H. Fu. The paper was written by J. Wu and W. Ai with the input of other authors. F. Luo co-supervised the whole project and gave constructive suggestions. All authors contributed to the scientific discussions.

## Competing interests

The authors declare no competing interests.
