## [Peer Review File · Nature Communications]

REVIEWER COMMENTS

Reviewer #1 (Remarks to the Author):

The authors presented their work on experimental and computational studies of anisotropic magnetoresistance in the beta phase of Ag₂Te topological material. While the results might be interesting, several important factors need to be carefully addressed.

On the experimental side, I think a more careful morphological and compositional analysis needs to be performed to be sure that we have the beta-Ag₂Te system. For example, the authors need to use the appropriate experimental approach to determine the purity of the sample. Chalcogens are known to induce native defects due to vacancy. This could play a huge role in the optoelectronic and electrophonic properties. Concerning what the authors refer to as “rough polycrystalline files”, it is not clear how this categorization is reached. Their samples are grown with CVD; I am more concerned about the purity rather than the surface morphology. For example, Refs. 31 and 32, and several others have characterized many Ag-based chalcogenide materials, including Ag₂Te. These previous studies also demonstrated large magnetoresistance. Could the authors show how their new findings contrast with these previous results?

In solving the effective Hamiltonian, it is not clear what the authors have done. Specifically, what served as input in obtaining the Green's function? Traditionally, for a material-specific mean-field approach, one would expect an effective Hamiltonian obtained from approaches such as wannier interpolation. If this was done for a generic single model, it would be beneficial to explain this in the manuscript. If the latter is the case, the question then is whether it could be more robust to directly use the phonon output from the phonon calculations and DFT electronic structure information in constructing the mean-field theory. This way, the underlying crystal symmetry and information are wholistically included in the mean-field solution. Also and most importantly, the low-energy effective Hamiltonian that was used in the analysis does not include any phonon information. How did the authors get the parameters such as the averaged electron-impurity and electron-ion interaction time used in the mean-field? There are many details and some are too simplistic (Bohr-like) to be able to fully capture or describe a topological system.

On style:

Firstly, I appreciate the scientific content and contributions of your paper. However, I have observed some instances in the manuscript where colloquial language and stylistic issues could be refined for greater clarity and professionalism.

For example, in the sentence: "Regarding the topological insulators, experimental observations of AMR and PHE were only faithfully and simultaneously achieved at low temperature up to now," the use of "faithfully" appears informal. I recommend replacing it with a more precise term, such as "reliably" or "consistently." This modification would enhance the precision of your statement.

Furthermore, the phrase "In a word," is somewhat informal and might benefit from a more formal transition. Consider replacing it with a phrase like "In summary" or "To summarize." In the sentence, "and will become invalid under the condition of a strong magnetic field," the phrase "and will become invalid" could be made more concise and precise. You may consider rephrasing it as "and becomes inapplicable in the presence of a strong magnetic field." There are many scenarios like this and I encourage the authors to carefully proofread the manuscript.

Lastly, I encourage you to review the manuscript for any other stylistic improvements that can enhance its overall clarity and professionalism. Paying attention to sentence structure, word choice, and transitions can contribute significantly to the readability and impact of your work.

Reviewer #2 (Remarks to the Author):

In this manuscript, Ai et al grew β -Ag₂Te nanoplates, studied their planar Hall effect (PHE), anisotropic magnetoresistance (AMR), as well as the possible origin of PHE and AMR, and devised the AMR magnetic sensors based on this compound. The current data suggest that β -Ag₂Te nanoplates are material candidates with novel PHE and AMR, and promising application potential in AMR sensors. To my opinion, this paper containing sufficient new results. However, several issues are yet to addressed.

(1)The abstract section should be carefully revised to make it more readable. For example, theoretical findings are suggested to be briefly presented. I am concerned that readers may not grasp the main points of this paper from the present abstract.

(2)In the introduction section, the authors should succinctly outline the theoretical background of planar Hall effects. Furthermore, the introduction section should be carefully organized, providing an in-depth insight into this area. The current content appears tedious to read.

(3)Also in the introduction section, the authors mentioned that the well-studied magnetic sensors are only capable of functioning in a narrow range of magnetic field (typically < 100 Gauss), and become functionally ineffective in the presence of strong magnetic field, thereby limiting the development of this field. Consequently, the authors concentrated their studies under strong magnetic fields primarily on the transport characteristic of β -Ag₂Te nanoplates. This appears somewhat far-fetched, considering that numerous materials exhibit linear and large magnetoresistance under strong magnetic field. From this standpoint, the article has forfeited its cutting-edge quality.

(4) In the results and discussion section, the authors presented the band structures and phonon dispersion relation. These findings have previously been reported and therefore appear duplicative.

(5) In Page 5, “The successful synthesis of β -Ag₂Te single-crystalline thin films greatly facilitates the investigation of its basic properties such as the PHE and AMR, which have never been reported in β -Ag₂Te before.” Such an assertion is deemed inappropriate. As far as know, AMR has been reported in references (Nano Lett. 2023, 23, 19, 9026–9033; Nano Lett. 2020, 20, 10, 7004–7010)

(6) The authors postulated that small effective masses and low sound velocities contribute towards β -Ag₂Te's PHE and AMR through theoretical analyses. I wonder whether or not the temperature dependence of β -Ag₂Te's PHE can be reproduced by utilizing their models. If not, how can the authors assert these points work for the PHE of β -Ag₂Te.

(7) Δ chiral pxy was not defined and introduced in the main text or Fig. 2c.

(8) How could the authors verify that the transport behaviors are primarily governed by the intrinsic electronic features, as their electron transport measurements were mainly performed at room temperature. I am intrigued to know if the thermally activated charges contribute to the transport.

Reviewer #3 (Remarks to the Author):

The authors reported the very large anisotropic magnetoresistance and the planar Hall effect at room temperature in Ag₂Te. Their calculations indicate that Ag₂Te is the topological insulator and they considered that the observed angular magnetoresistance originates from the Dirac-type surface states on the basis of their theoretical model. Furthermore, they performed the nice demonstration of the angle sensitive magnetic sensor by preparing the micro-scale Wheatstone bridge device of Ag₂Te. I think the experimental results are very interesting and the manuscript is well written. However, I have following questions.

The authors consider the tilting of the Dirac cone in a magnetic field is the possible origin of the anisotropic conduction. However, I think the resistivity in the present sample is very low for the surface conduction. Can the authors exclude the possibility of the bulk conduction that is widely seen in various topological materials by the tiny amounts of impurities or defects? Could you show the temperature dependence of the longitudinal resistivity R_{xx} down to 2 K? Are there some characteristic features of the surface conduction? Could you estimate the carrier density?

Could you show the angular magnetoresistance in various temperatures down to 2 K? The amplitude of the AMR ratio decreases with decreasing temperature as seen in the planar Hall effect?

High mobility semimetal NbAs also shows the large anisotropic magnetoresistance $\sim 200\%$ at room temperature at 9 T. For example, *Journal of Physics: Condensed Matter* 27 152201 (2015). What is the advantage in the present system as the magnetic sensor by comparison with those high mobility semimetals?

Reviewer #1 (Remarks to the Author):

The authors presented their work on experimental and computational studies of anisotropic magnetoresistance in the beta phase of Ag₂Te topological material. While the results might be interesting, several important factors need to be carefully addressed.

Authors' response

We highly appreciate the positive evaluations and professional suggestions of the referee, which greatly improve the quality of our manuscript. The point-by-point response are listed as follows.

1. On the experimental side, I think a more careful morphological and compositional analysis needs to be performed to be sure that we have the beta-Ag₂Te system. For example, the authors need to use the appropriate experimental approach to determine the purity of the sample. Chalcogens are known to induce native defects due to vacancy. This could play a huge role in the optoelectronic and electrophonic properties.

Authors' response

Thanks for the reviewer's professional question.

First, we totally agree with the reviewer that "*the authors need to use the appropriate experimental approach to determine the purity of the sample*". In our previous version, we have used a series of experimental tools to analyze the crystalline phase and elemental compositions of as-synthesized samples, including the SEM EDX mapping (averaging the value on 10 samples, Fig. S7 and Table S1), TEM imaging along two zone axes (Figs. 1h-i, and Fig. S5), atomic-resolved EDS mapping (Figs. 1j-k) and XPS measurements (Fig. S6). All of them can be assigned to the beta phase of Ag₂Te with a nearly perfect atomic ratio of Ag: Te = ~2 :1. However, we should admit that the possible stoichiometric deviations induced by trace amount of defects, such as 0.001%~0.1%, are almost undetectable by EDX with a detection limit of ~1% and the existing other quantitative tools. Fortunately, as pointed out by the reviewer, trace amount of defects can greatly influence the electrical properties of a material, such as carrier density and conductivity. To this end, one can extract the carrier density of a material by Hall measurement to indirectly reflect the amount of defects in it. As shown in Fig. R1, the CVD-grown β -Ag₂Te is *n*-type, and has a low carrier density of $\sim 1.0 \times 10^{18} \text{ cm}^{-2}$ at room temperature, which is much lower than the value ($10^{19} \sim 10^{20} \text{ cm}^{-3}$) of well-known Bi₂Se₃ topological insulator. Let's suppose one defect site donates one electron, thus the defects density for the carrier density of $\sim 10^{18} \text{ cm}^{-3}$ can be estimated as 0.01%~0.001%. Such trace amount of defects are almost impossible to accurately determine by all the existing experimental tools for elemental analysis.

Fig. R1 | The extracted Hall mobility and carrier density of CVD-grown β -Ag₂Te as a function of temperature. The room-temperature carrier density can be as low as $1.0 \times 10^{18} \text{ cm}^{-3}$, and Hall mobility can be as high as $\sim 8 \times 10^3 \text{ cm}^2 \text{ V}^{-1} \text{ s}^{-1}$ at room temperature.

Second, to investigate the potential microscopic conduction mechanism in β -Ag₂Te, we performed DFT calculations to see the formation energies and dopant type for four possible different defects. As shown in Figs. R2a-d, the Te vacancy (V_{Te}) and Ag_{Te} antisite defects act as the electron donor (n-type) in β -Ag₂Te, while the Ag vacancy (V_{Ag}) and Te_{Ag} antisite defects are the acceptor dopants (p-type) instead. That is to say, the system tends to demonstrate n-type behavior in Ag-rich environments, while exhibiting p-type characteristics in Ag-poor conditions. This behavior is consistent with the previous experimental results, in which Ag_{2+x}Te and Ag_{2-x}Te are n-type and p-type, respectively (see H. S. Schnyders, et al. *Appl. Phys. Lett.* 76, 1710 (2000)). Furthermore, our investigation revealed that the Ag_{Te} substitutional defect has the smallest formation energy, thus acting as the preferred n-type dopant and matching well the experimental results of Hall measurements (slightly n doped in CVD-grown β -Ag₂Te, Fig. R1).

In our revision, we added the temperature-dependent carrier density and Hall mobility (Fig. R1) and calculated band structures and formation energies for β -Ag₂Te with possible defects (Fig. R2) in the supporting information as Fig. S8 and Fig. S9, and clearly point out the possible existence of stoichiometric deviations induced by trace amount of defects in the main text.

Fig. R2 | The band structures (a-d) and formation energies (e) for β -Ag₂Te supercell with four possible defects by DFT calculations. Formation energies for four types of defects, encompassing two vacancies (V_{Ag} and V_{Te}) and two substitutional defects (Ag_{Te} and Te_{Ag}), in a $2 \times 2 \times 2$ β -Ag₂Te supercell considering SOC. In the case of the Ag_{Te} substitutional defect, one Ag atom substitutes for one Te atom in the supercell. Conversely, for Te_{Ag} , one Te atom substitutes for one Ag atom in the supercell. The formation energy is defined as $E_f = (E_{defect} + E_{atom}) - E_{perfect}$ for vacancy defects and $E_f = (E_{defect} + E_{substituted\ atom} - E_{replacing\ atom}) - E_{perfect}$ for substitutional defects. E_{defect} and $E_{perfect}$ represent the total energy of the supercell with a defect and the perfect supercell, respectively. E_{atom} corresponds to the energy of an individual atom that has been removed to create a vacancy. $E_{substituted\ atom}$ and $E_{replacing\ atom}$ correspond to the energy of the atom being substituted and the energy of the atom replacing it, respectively.

2. Concerning what the authors refer to as “rough polycrystalline films”, it is not clear how this categorization is reached. Their samples are grown with CVD; I am more concerned about the purity rather than the surface morphology.

Authors’ response

Thanks for the referee's question.

Here the "rough polycrystalline films" refer to polycrystalline β -Ag₂Te synthesized by modified Bridgman method usually have a large surface roughness. Notably, unlike other topological insulators (such as Bi₂Se₃), it is quite challenging to synthesize the bulk single crystals of β -Ag₂Te. As a result, previous works regarding the large linear magnetoresistance in β -Ag₂Te, such as Ref. 31 and 32, were all based on the polycrystalline films with a grain size of >500 nm, rather than single crystalline films. To our knowledge, the segregation and non-uniform distribution of elements may occur in the large polycrystalline ingot of β -Ag₂Te, which may hinder us to further investigate the intrinsic properties of β -Ag₂Te. Compared to the previously reported polycrystalline films, the CVD-grown β -Ag₂Te nanosheets are single crystalline and have an ultra-smooth surface, which are preferable for the fundamental research and electronic applications.

To avoid possible confusion, we removed the word "rough" in the revision.

3. For example, Refs. 31 and 32, and several others have characterized many Ag-based chalcogenide materials, including Ag₂Te. These previous studies also demonstrated large magnetoresistance. Could the authors show how their new findings contrast with these previous results?

Authors' response

Thanks for the referee's professional question. If we understand correctly, this question contains two parts.

1) What's the conceptual difference between magnetoresistance and anisotropic magnetoresistance.

To our knowledge, numerous high-mobility materials can show very large **out-of-plane** magnetoresistance under strong magnetic field. Of course, those materials can be functionalized into high-field magnetometers to detect the strength of magnetic field. However, large magnetoresistance only means sensitive resistance change under a magnetic field with a fixed angle (perpendicular to the device plane). It does not mean the capability to detect the angle between the current and applied magnetic field with high sensitivity, which is the main function of AMR/GMR sensors. Moreover, we should emphasize that the AMR/GMR effect that used to fabricate AMR/GMR sensor has a specific meaning and strict definition, which is only related to the resistance difference by changing the angle between the current and applied **in-plane** magnetic field (Fig. R3a), rather than magnetic field with arbitrary angle. To this end, the main function of AMR/GMR sensor is to **detect the in-plane angle change** (Fig. R3b), rather than out-of-plane one. In other words, it is the large in-plane AMR value matters for the fabrication of AMR angle sensors, rather than the large out-of-plane magnetoresistance. Actually, it is quite normal to observe the change of magnetoresistance when the applied magnetic field is rotated from out of plane to in plane, since the out-of-plane component of magnetic field changes gradually. Besides, large out-of-plane magnetoresistance has nothing to do with large in-plane AMR value at all, which usually depends on the in-plane anisotropic scattering of conductive electrons.

Fig. R3 | **a**, Schematic diagram of measuring the in-plane anisotropic resistance of a material. **b**, Illustration of the fabricated AMR angle sensor setup, in which the magnet is fixed on an in-plane rotator.

2) *How our new findings contrast with these previous results*

Just as pointed out by the reviewer, β -Ag₂Te is an old material and was found to present large and linear **out-of-plane** magnetoresistance in its polycrystalline form at room temperature (Xu, R. et al. *Nature* 309, 57-60 (1997); Hu, J. et al. *Phys. Rev. Lett.* 95, 186603 (2005)). Up to now, there is no reports on the planar Hall effect and corresponding in-plane AMR of β -Ag₂Te, which has the potential to fabricate AMR angle sensors. In this work, we presented the first observation of planar Hall effect and in-plane anisotropic resistance in CVD-grown β -Ag₂Te, which showed an unexpected, record-high AMR ratio of -39 % and giant planar Hall effect (520 $\mu\Omega\cdot\text{cm}$) at room temperature, both of which are more than one order of magnitude higher than all the previously reported values. Besides, a theoretical model was proposed to illustrate its possible origin. More importantly, based on a single CVD-grown β -Ag₂Te nanoplate, we fabricated the first topological-insulator based AMR sensors with a small device size, showing ultrahigh output voltages (240 mV/V), high angle sensitivity (4.2 mV/V/deg) and small angle error ($< 1^\circ$).

4. In solving the effective Hamiltonian, it is not clear what the authors have done. Specifically, what served as input in obtaining the Green's function? Traditionally, for a material-specific mean-field approach, one would expect an effective Hamiltonian obtained from approaches such as wannier interpolation. If this was done for a generic single model, it would be beneficial to explain this in the manuscript. If the latter is the case, the question then is whether it could be more robust to directly use the phonon output from the phonon calculations and DFT electronic structure information in constructing the mean-field theory. This way, the underlying crystal symmetry and information are wholistically included in the mean-field solution.

Authors' response

We highly appreciate the reviewer's professional suggestion. The Referee might have misunderstood the mean-field approximation we mean here. We did not solve the effective Hamiltonian from a material-specific mean-field theory, but used the mean-field approximation to truncate the iterative equations for the Green's functions. The Green's function here is derived

directly from the well-accepted $\mathbf{k} \cdot \mathbf{p}$ effective Hamiltonian for the surface states of 3D topological insulators, where the phonon reservoir is treated as a background of the electrons. As a result, the electron-phonon interaction enters the system via the self-energy of the Green's functions. This treatment has been widely-adopted in literatures [*Phys. Rev. Lett.* 125, 206601 (2020); *Nat. Commun.* 8, 2267(2017); *Phys. Rev. B* 88, 075432 (2013); *Phys. Rev. B* 82, 195403 (2010)].

Since the electron-phonon interaction term in the Hamiltonian involves product of three operators, e.g., $D_q(b_q + b_{-q}^\dagger)c_{k+q}^\dagger c_k$, the equations of motion for the Green's functions contain

higher-order Green's functions, such as $\left\langle\left\langle c_{k-q}c_{l-q}^\dagger c_l \mid c_{k'}^\dagger \right\rangle\right\rangle_\epsilon$, making the iterative equations not

closed, please see the revised SI. To close the iterative equations, we truncate the higher-order Green's functions following the standard mean-field approximation method by contracting the

operator pairs, for example, $\left\langle\left\langle c_{k-q}c_{l-q}^\dagger c_l \mid c_{k'}^\dagger \right\rangle\right\rangle_\epsilon = \langle c_{k-q}c_{l-q}^\dagger \rangle \langle\langle c_l \mid c_{k'}^\dagger \rangle\rangle_\epsilon - \langle c_{k-q}c_l \rangle \left\langle\left\langle c_{l-q}^\dagger \mid \right.\right.$

$\left. c_{k'}^\dagger \right\rangle\rangle_\epsilon + \langle c_{l-q}^\dagger c_l \rangle \langle\langle c_{k-q} \mid c_{k'}^\dagger \rangle\rangle_\epsilon$, as extensively-adopted in references [*Phys. Rev.* 188, 874 (1969);

Phys. Rev. B 84, 165105 (2011); *Phys. Rev. B* 106, 195134(2022)]. With the mean-field approximation mentioned above, we can obtain an analytical expression for the Green's function of the surface electrons.

In the revised manuscript, we have explained the mean-field approximation adopted and the detailed treating processes are added to the revised SI.

5. Also and most importantly, the low-energy effective Hamiltonian that was used in the analysis does not include any phonon information. How did the authors get the parameters such as the averaged electron-impurity and electron-ion interaction time used in the mean-field? There are many details and some are too simplistic (Bohr-like) to be able to fully capture or describe a topological system.

Authors' response

As mentioned above, the low-energy effective Hamiltonian we used has been widely accepted in literatures and the phonon information has been included in the electron self-energy. The averaged electron-impurity and electron-ion interaction time can be extracted from the self-energy. It should be noted that our model is a qualitative analysis, rather than a quantitative tool, to see how the PHE amplitude changes with the parameters and thus help us to understand and figure out the possible origin of large room-temperature AMR ratio and PHE in beta-Ag₂Te system. Our theoretical results are qualitatively consistent with the experimental measurements, which enable the investigation of the underlying physics.

On style:

Firstly, I appreciate the scientific content and contributions of your paper. However, I have observed some instances in the manuscript where colloquial language and stylistic issues could be refined for greater clarity and professionalism.

For example, in the sentence: "Regarding the topological insulators, experimental observations of AMR and PHE were only faithfully and simultaneously achieved at low temperature up to now," the use of "faithfully" appears informal. I recommend replacing it with a more precise term, such as "reliably" or "consistently." This modification would enhance the precision of your statement.

Furthermore, the phrase "In a word," is somewhat informal and might benefit from a more formal transition. Consider replacing it with a phrase like "In summary" or "To summarize." In the sentence, "and will become invalid under the condition of a strong magnetic field," the phrase "and will become invalid" could be made more concise and precise. You may consider rephrasing it as "and becomes inapplicable in the presence of a strong magnetic field." There are many scenarios like this and I encourage the authors to carefully proofread the manuscript.

Lastly, I encourage you to review the manuscript for any other stylistic improvements that can enhance its overall clarity and professionalism. Paying attention to sentence structure, word choice, and transitions can contribute significantly to the readability and impact of your work.

Authors' response

Thanks for the referee's kindness to point out colloquial language and stylistic issues in our paper. In our revision, we have adopted all your suggestions to make our expression more professional, such as the word "faithfully" was replaced by "reliably", "in a word" was replaced by "in summary", "will become invalid" was replaced by "and becomes inapplicable in the presence of a strong magnetic field". In addition, we have double-checked the whole paper thoroughly to improve readability.

Reviewer #2 (Remarks to the Author):

In this manuscript, Ai et al grew β -Ag₂Te nanoplates, studied their planar Hall effect (PHE), anisotropic magnetoresistance (AMR), as well as the possible origin of PHE and AMR, and devised the AMR magnetic sensors based on this compound. The current data suggest that β -Ag₂Te nanoplates are material candidates with novel PHE and AMR, and promising application potential in AMR sensors. To my opinion, this paper containing sufficient new results. However, several issues are yet to be addressed.

Authors' response

We highly appreciate the positive evaluations and professional suggestions of the referee. Following the reviewer's suggestion, we have carefully polished our language and writing styles to make our paper more readable. Please see below for the point-by-point response.

(1) The abstract section should be carefully revised to make it more readable. For example, theoretical findings are suggested to be briefly presented. I am concerned that readers may not

grasp the main points of this paper from the present abstract.

Authors' response

Thanks for the reviewer's kind suggestion. The brief introduction about theoretical finding has been added to the abstract section, and we have revised our manuscript carefully to increase readability.

(2) In the introduction section, the authors should succinctly outline the theoretical background of planar Hall effects. Furthermore, the introduction section should be carefully organized, providing an in-depth insight into this area. The current content appears tedious to read.

Authors' response

Thanks for the reviewer's suggestion. In our revision, we added several sentences regarding the theoretical background of planar Hall effect to the introduction section, and made appropriate modifications to the overall introduction to improve the readability.

(3) Also in the introduction section, the authors mentioned that the well-studied magnetic sensors are only capable of functioning in a narrow range of magnetic field (typically < 100 Gauss), and become functionally ineffective in the presence of strong magnetic field, thereby limiting the development of this field. Consequently, the authors concentrated their studies under strong magnetic fields primarily on the transport characteristic of β -Ag₂Te nanoplates. This appears somewhat far-fetched, considering that numerous materials exhibit linear and large magnetoresistance under strong magnetic field. From this standpoint, the article has forfeited its cutting-edge quality.

Authors' response

Thanks for the reviewer's kind comment.

First, we should admit the possible imprecise expression of magnetic sensor in our previous version may cause some misleading to the referee. Yes, just as pointed out by the reviewer, numerous materials exhibit linear and large magnetoresistance under strong **out-of-plane** magnetic field. Of course, those materials can also be functionalized into high-field magnetometers to detect the strength of magnetic field. However, large magnetoresistance only means sensitive resistance change under a magnetic field with a fixed angle (perpendicular to the device plane). **It does not mean the capability to detect the angle** between the current and applied magnetic field with high sensitivity, which is the main function of AMR/GMR sensors.

Second, we should emphasize that the AMR/GMR effect that used to fabricate AMR/GMR sensor has a specific meaning and strict definition, which is only related to the resistance difference by changing the angle between the current and applied **in-plane** magnetic field (Fig. R4a), rather than magnetic field with arbitrary angle. To this end, the main function of AMR/GMR sensor is to **detect the in-plane angle change** (Fig. R4b), rather than out-of-plane one. In other words, it is the large in-plane AMR value matters for the fabrication of AMR sensor, rather than the large out-of-plane magnetoresistance. In fact, large out-of-plane magnetoresistance has nothing to do with large in-plane AMR value at all, which usually depends on the in-plane anisotropic scattering of conductive electrons.

Fig. R4 | **a**, Schematic diagram of measuring the in-plane anisotropic resistance of a material. **b**, Illustration of the fabricated AMR angle sensor setup, in which the magnet is fixed on an in-plane rotator.

To avoid the possible misleading and confusion, we more emphasized the “in-plane magnetic field” and the function of in-plane “angle detection” for AMR/GMR sensor” in our revision.

(4) In the results and discussion section, the authors presented the band structures and phonon dispersion relation. These findings have previously been reported and therefore appear duplicative.

Authors’ response

Thanks a lot for the reviewer's kind remind. In fact, we performed the independent DFT calculations on the band structure and phonon dispersion relation in this work, and gave the similar results. In our revision, we have changed the dispersion pathway of Brillouin zone and marked out the orbit contribution of Ag and Te to avoid duplication of the previous report, as shown in Fig. R5.

Fig. R5 | New version of the calculated band structure (a) and phonon dispersion (b) of β -Ag₂Te.

(5) In Page 5, “The successful synthesis of β -Ag₂Te single-crystalline thin films greatly facilitates

the investigation of its basic properties such as the PHE and AMR, which have never been reported in β -Ag₂Te before.” Such an assertion is deemed inappropriate. As far as know, AMR has been reported in references (Nano Lett. 2023, 23, 19, 9026–9033; Nano Lett. 2020, 20, 10, 7004–7010)

Authors’ response

Thanks for the referee's kind remind. To our knowledge, the references (Nano Lett. 2023, 23, 19, 9026–9033; Nano Lett. 2020, 20, 10, 7004–7010) only investigated the angle-dependent R_{xx} of Ag₂Te with the magnetic field rotated **from out of plane to in plane** at low temperature (2 K). In fact, it is quite normal to observe the change of magnetoresistance when the applied magnetic field is rotated from out of plane to in plane, since the out-of-plane component of magnetic field changes gradually. However, as I mentioned in question 3#, such kind of magnetoresistance change is NOT the so-called AMR at all. The AMR is only related to the resistance difference by changing the angle between the current and applied **in-plane** magnetic field, rather than magnetic field with arbitrary angle. To this end, the PHE and AMR effects are indeed not reported before. To avoid possible confusion, the word “AMR” was changed to “in-plane AMR” in our revision.

(6) The authors postulated that small effective masses and low sound velocities contribute towards β -Ag₂Te's PHE and AMR through theoretical analyses. I wonder whether or not the temperature dependence of β -Ag₂Te's PHE can be reproduced by utilizing their models. If not, how can the authors assert these points work for the PHE of β -Ag₂Te.

Authors’ response

Thanks for the referee’s professional question. Yes, the unconventional non-monotonic temperature of PHE amplitude can be obtained based on our model, which fits well with the one observed in β -Ag₂Te and other topological insulators. Moreover, with decreasing the sound velocities and effective mass, the peak temperature (T_p) gradually moves to a higher temperature, accompanied by a significantly enhanced PHE amplitude (Fig. R6). It is worth noting that our model is a qualitative analysis, rather than a quantitative tool, to see how the PHE amplitude changes with the parameters and thus help us to understand and figure out the possible origin of large room-temperature AMR ratio and PHE in beta-Ag₂Te system.

In our revision, Fig. R6 was added as new Fig. S15 in supporting information.

Fig. R6 | a, The PHE amplitude $\Delta\rho/\rho_0$ vs temperature for varied sound velocities (v_s) and effective masses

(m_{eff}). **b**, The differential form of $\Delta\rho/\rho_0$ as a function of temperature, indicating that the peak temperature (T_p) gradually moves to a higher temperature with decreasing the v_s .

(7) $\Delta^{\text{chiral}}\rho_{xy}$ was not defined and introduced in the main text or Fig. 2c.

Authors' response

Thanks for the referee's kindness to point out the tiny error in our paper. The " $\Delta^{\text{chiral}}\rho_{xy}$ " was replaced by $\Delta\rho_{xy}$, which was defined as $\rho_{xy}(\varphi=145^\circ) - \rho_{xy}(\varphi=45^\circ)$ in the figure caption of Fig. 2c.

(8) How could the authors verify that the transport behaviors are primarily governed by the intrinsic electronic features, as their electron transport measurements were mainly performed at room temperature. I am intrigued to know if the thermally activated charges contribute to the transport.

Authors' response

We thank the reviewer for raising this inspiring open question. To our knowledge, with increasing the temperature, the hopping conduction induced by non-intrinsic electronic states, such as disorders and localized states, may dominate the charge transport in a low-quality sample. Nevertheless, the CVD-grown β - Ag_2Te nanoplate has a very high room-temperature Hall mobility of $\sim 8000 \text{ cm}^2 \text{ V}^{-1} \text{ s}^{-1}$, indicating the ignorable imperfection of crystal lattice. More importantly, such kind of thermally activated charges from non-intrinsic electronic states will undergo an isotropic scattering, rather than anisotropic scattering, in the presence of magnetic field, thus will not contribute to planar Hall effect and anisotropic magnetoresistance.

Reviewer #3 (Remarks to the Author):

The authors reported the very large anisotropic magnetoresistance and the planar Hall effect at room temperature in Ag_2Te . Their calculations indicate that Ag_2Te is the topological insulator and they considered that the observed angular magnetoresistance originates from the Dirac-type surface states on the basis of their theoretical model. Furthermore, they performed the nice demonstration of the angle sensitive magnetic sensor by preparing the micro-scale Wheatstone bridge device of Ag_2Te . I think the experimental results are very interesting and the manuscript is well written. However, I have following questions.

Authors' response

We really appreciate the reviewer's positive evaluations and constructive suggestions on our work, which really help us to greatly improve the quality of our work. Following the reviewer's suggestion, we have provided extra data. Please see below for our point-by-point response.

(1) The authors consider the tilting of the Dirac cone in a magnetic field is the possible origin of the anisotropic conduction. However, I think the resistivity in the present sample is very low for the surface conduction. Can the authors exclude the possibility of the bulk conduction that is widely seen in various topological materials by the tiny amounts of impurities or defects? Could you show the temperature dependence of the longitudinal resistivity R_{xx} down to 2 K? Are there some characteristic features of the surface conduction? Could you estimate the carrier density?

Authors' response

We appreciate the reviewer's professional questions. The temperature dependence of the longitudinal resistivity (R_{xx}), Hall mobility and carrier density were shown in Fig. R7. The R_{xx} - T curve is similar to the previous one observed in β -Ag₂Te. Besides, unlike the well-known Bi₂Se₃ topological insulator with a high carrier density ($>10^{19}$ cm⁻³), the CVD-grown β -Ag₂Te has a relatively low carrier density of $\sim 1.0 \times 10^{18}$ cm⁻³ at room temperature, which gradually decreased to $\sim 4 \times 10^{17}$ cm⁻³ at 5 K. It is worth nothing that β -Ag₂Te has a very high room-temperature Hall mobility of $\sim 8 \times 10^3$ cm² V⁻¹ s⁻¹, which accounts for its low resistivity even when it has a low carrier density. Furthermore, one can get the relationship between Fermi level and carrier density by calculating a material's electronic states near the Dirac point. For the case of β -Ag₂Te, such a low carrier density of $\sim 10^{18}$ cm⁻³ corresponds to a Fermi level of ~ 40 meV above the Dirac point (see *Nat. Mater.* 22, 860 (2023), which is near the conduction band minimum of β -Ag₂Te.

Moreover, to investigate whether the bulk states of topological insulator can induce planar Hall effect and its characteristics, we performed theoretical calculation for the bulk Hamiltonian of 3D topological insulators by using the Boltzmann theory. As shown in Fig. R8a, the bulk conduction of the topological insulator will also induce a planar Hall effect because of the Berry curvature, which is similar to the chirality anomaly in Dirac semimetal. However, the bulk states induced PHE will show a conventional temperature dependence on the PHE amplitude instead (Fig. R8b), namely having a high PHE amplitude at low temperature but vanishing to near zero at room temperature. Apparently, it is totally different from the one observed in our case, indicating that our large room-temperature PHE is not primarily governed by bulk states. In all, considering the low carrier density and unconventional temperature dependence on PHE amplitude, we believe the surface states of β -Ag₂Te still dominates the charge transport at room temperature, even though a small amount of bulk states will inevitably contribute to the charge conduction.

In our revised version, we added the temperature-dependent longitudinal resistivity and carrier density (Fig. R7) and the calculated bulk states induced PHE (Fig. R8) in the Supporting information (Fig. S8 and Fig. S16).

Fig. R7 | a, Longitudinal resistance as a function of temperature ranging from 320 K to 4 K, **b**,

The extracted Hall mobility and carrier density as a function of temperature.

Fig. R8 | The calculated angle (a) and temperature (b) dependence of PHE amplitude induced by the bulk states of the topological insulator, showing a monotonic decrease upon warming up.

(2) Could you show the angular magnetoresistance in various temperatures down to 2 K? The amplitude of the AMR ratio decreases with decreasing temperature as seen in the planar Hall effect?

Authors' response

Thanks for reviewer's professional question. Following the reviewer's suggestion, we provided temperature-dependent angular magnetoresistance and corresponding AMR amplitude. As shown in **Fig. R9**, the amplitude of AMR also exhibits an unconventional increase with temperature increasing up to 300 K, which is similar to the tendency observed in planar Hall effect. In our revision, the Fig. R9 is added as new **Fig. S13** in supporting information.

Fig. R9 | a, Temperature-dependent AMR measurements at varied temperatures from 320 to 4 K. The in-plane magnetic field is kept constant as 9 T. **b**, The extracted AMR amplitude as a function of temperature on the basis of **a**. Here, the AMR amplitude is defined as $R_{xx,\perp} - R_{xx,\parallel}$.

(3) High mobility semimetal NbAs also shows the large anisotropic magnetoresistance ~200% at room temperature at 9 T. For example, Journal of Physics: Condensed Matter 27 152201 (2015).

What is the advantage in the present system as the magnetic sensor by comparison with those high mobility semimetals?

Authors' response

Thanks for the referee's professional question.

First, let me make it clear that the semimetal NbAs in this literature [*Journal of Physics: Condensed Matter* 27 152201 (2015)] presents a large **out-of-plane** magnetoresistance (~200%), rather than anisotropic magnetoresistance at room temperature and 9 T. In fact, numerous high-mobility materials can show very large magnetoresistance under strong **out-of-plane** magnetic field. Of course, those materials can also be functionalized into high-field magnetometers to detect the strength of magnetic field. However, large magnetoresistance only means sensitive resistance change under a magnetic field with a fixed angle (perpendicular to the device plane). It does not mean the capability to detect the angle between the current and applied magnetic field with high sensitivity, which is the main function of AMR/GMR sensors.

Second, we should emphasize that the AMR/GMR effect that used to fabricate AMR/GMR sensor has a specific meaning and strict definition, which is only related to the resistance difference by changing the angle between the current and applied **in-plane** magnetic field (Fig. R10), rather than magnetic field with arbitrary angle. To this end, the main function of AMR/GMR sensor is to **detect the in-plane angle change**, rather than out-of-plane one. In other words, it is the large in-plane AMR value matters for the fabrication of AMR sensor, rather than the large out-of-plane magnetoresistance. In fact, it is quite normal to observe the change of magnetoresistance when the applied magnetic field is rotated from out of plane to in plane, since the out-of-plane component of magnetic field changes gradually. Besides, large out-of-plane magnetoresistance has nothing to do with large in-plane AMR value at all, which usually depends on the in-plane anisotropic scattering of conductive electrons.

To avoid the possible misleading and confusion, we more emphasized the “in-plane magnetic field” and the function of in-plane “angle detection” for AMR/GMR sensor” in our revision.

Fig. R10 a, Schematic diagram of measuring the in-plane anisotropic resistance of a material. **b**,

Illustration of the fabricated AMR angle sensor setup, in which the magnet is fixed on an in-plane rotator.

REVIEWER COMMENTS

Reviewer #1 (Remarks to the Author):

The authors have addressed most of the concerns and queries raised in my previous comments, as well as those posed by fellow reviewers. I believe that this has resulted in substantial improvements to the readability of the manuscript.

I firmly believe that the newfound insights generated through the experimental measurement of the planar Hall effect and the corresponding in-plane anisotropic magnetoresistance (AMR) in β -Ag₂Te hold significant promise. These insights could prove instrumental in advancing ongoing research efforts related to the development and fabrication of AMR-based sensors.

Reviewer #2 (Remarks to the Author):

The authors have thoroughly revised their manuscript. Despite this, I persist in my questioning of their proposed mechanism's capability to reproduce PHE and AMR in β -Ag₂Te, mainly due to their exciting and unique findings. I note that in the band structures there exists a tiny gap ~ 35 meV that close to the thermal energy at 300 K (26 meV). How can the authors rule out the contribution from bulk states to their transport? Their claim should be cautious. If their hypothesis does indeed explain the observed phenomena, it could provide novel insights into future advancement in the field.

Additionally, notable minor issues persist:

(1) in this sentence “a record-high AMR ratio of -39 % and giant planar Hall effect (520 $\mu\Omega\cdot\text{cm}$) were observed at room temperature”, the authors should clarify the measurement conditions (under 9 T);

(2) “To investigate the band structure and phonon dispersion of β -Ag₂Te, we performed density functional theory (DFT) calculations (Figs. 1b-d). β -Ag₂Te possess a small bulk band gap of ~ 35 meV, and its nontrivial topological nature with a nontrivial invariant $Z_2 = 1$ is confirmed by analyzing the parity of occupied Bloch wave functions at the time-reversal invariant points in Brillouin zone, which undoubtedly support gapless Dirac-type surface states inside the band gap (Fig. 1b).” It is debatable whether the authors verified the value of Z_2 independently. If not, it is recommended that this sentence be eliminated or a source referenced.

Reviewer #3 (Remarks to the Author):

I found that the authors nicely revised the manuscript and included the new data requested by the referees.

Although I think that the electrical resistivity in the conventional high mobility semimetals is very sensitive to the magnetic field direction when the angle is changed from the out-of plane to the current direction (anisotropy is comparable with the magnetoresistance ratio), I understand the mechanism in the present system is totally different and the results are novel.

I recommend the publication of the paper.

Reviewer #1 (Remarks to the Author):

The authors have addressed most of the concerns and queries raised in my previous comments, as well as those posed by fellow reviewers. I believe that this has resulted in substantial improvements to the readability of the manuscript.

I firmly believe that the newfound insights generated through the experimental measurement of the planar Hall effect and the corresponding in-plane anisotropic magnetoresistance (AMR) in β -Ag₂Te hold significant promise. These insights could prove instrumental in advancing ongoing research efforts related to the development and fabrication of AMR-based sensors.

Authors' response

We highly appreciate the reviewer's high appraisal on the achievement and significance of our work, which greatly encourage us to devote future research enthusiasm in this material and its derivative. Thanks again for your constructive suggestions and inspiring questions raised during our previous revision.

Reviewer #2 (Remarks to the Author):

The authors have thoroughly revised their manuscript. Despite this, I persist in my questioning of their proposed mechanism's capability to reproduce PHE and AMR in β -Ag₂Te, mainly due to their exciting and unique findings. I note that in the band structures there exists a tiny gap ~ 35 meV that close to the thermal energy at 300 K (26 meV). How can the authors rule out the contribution from bulk states to their transport? Their claim should be cautious. If their hypothesis does indeed explain the observed phenomena, it could provide novel insights into future advancement in the field.

Authors' response

First, we sincerely thank the reviewer for his/her kindness and great patience to point out the possible existence of imprecise expression in our paper. If we understand correctly, the reviewer suggests us to adopt a more conservative way when talk about the mechanism we proposed. Just as pointed out by the reviewer, we observed very exciting and unique findings regarding the giant room-temperature PHE and in-plane AMR in CVD-grown β -Ag₂Te, and proposed a theoretical model to investigate its possible origin. Moreover, based on the theoretical model, we indeed can deduce the giant room-temperature AMR value and abnormal non-monotonic temperature dependence on PHE amplitude, both of which fit well with our experimental results. Considering the consistency between our proposed theoretical model and experimental results, we, therefore, regard it as one of the possible underlying mechanisms. However, just as reminded by the reviewer, we should admit that we cannot exclude that other models can also give the similarly consistent results with the experimental data. To be more rigorous, we pointed it out clearly in the main text as follows.

"It should be noted that our model is a qualitative analysis, rather than a quantitative tool, to see how the PHE amplitude changes with the parameters and thus help us to understand and figure out the possible origin of large room-temperature AMR ratio and PHE in β -Ag₂Te system. Our

theoretical results are qualitatively consistent with the experimental measurements, which enable the investigation of the underlying physics. However, we can not exclude that other models can give the similarly consistent results, which need further investigations in the future.”

Fig. R1 | The calculated angle (a) and temperature (b) dependence of PHE amplitude induced by the bulk states of the topological insulator, showing a monotonic decrease upon warming up.

Second, regarding *the contribution from the bulk states to the electrical transport*, the 3rd reviewer asked the same open question in his/her last-round remarks. In fact, we have taken this open question seriously by performing extra theoretical calculations for the bulk Hamiltonian of 3D topological insulators by using the Boltzmann theory. As shown in Fig. R1a, the bulk conduction of the topological insulator will also induce a planar Hall effect because of the Berry curvature, which is similar to the chiral anomaly in Dirac semimetal (see *Phys. Rev. Lett.* 2017, 119, 176804). However, the bulk states induced PHE will show a conventional temperature dependence on the PHE amplitude instead (Fig. R1b), namely having a high PHE amplitude at low temperature but vanishing to near zero at room temperature. Apparently, it is totally different from the one observed in our case, indicating that our large room-temperature PHE is not primarily governed by bulk states. Besides, unlike the well-known Bi₂Se₃ topological insulator with a high carrier density (>10¹⁹ cm⁻³), the CVD-grown β-Ag₂Te has a relatively low carrier density of ~1.0*10¹⁸ cm⁻³ at room temperature, which gradually decreased to ~4*10¹⁷ cm⁻³ at 5 K. Furthermore, one can get the relationship between Fermi level and carrier density by calculating a material’s electronic states near the Dirac point. For the case of β-Ag₂Te, such a low carrier density of ~10¹⁸ cm⁻³ corresponds to a Fermi level of ~40 meV above the Dirac point (see *Nat. Mater.* 22, 860 (2023), which is near the conduction band minimum of β-Ag₂Te.

In our revision, to be more rigorous, we clearly point it out in the main text that *“considering the low carrier density and unconventional temperature dependence on PHE amplitude, we believe the surface states of β-Ag₂Te still dominate the charge transport at room temperature, even though a small amount of bulk states will inevitably contribute to the charge conduction.”*

Additionally, notable minor issues persist:

(1) in this sentence “a record-high AMR ratio of -39 % and giant planar Hall effect (520 μΩ.cm) were observed at room temperature”, the authors should clarify the measurement conditions

(under 9 T);

Authors' response

Thanks for the referee's kindness to point out the expression imprecision in our paper. In our revision, the measurement condition (under 9 T) has been added to the as-mentioned sentence.

(2) "To investigate the band structure and phonon dispersion of β -Ag₂Te, we performed density functional theory (DFT) calculations (Figs. 1b-d). β -Ag₂Te possess a small bulk band gap of ~ 35 meV, and its nontrivial topological nature with a nontrivial invariant $Z_2 = 1$ is confirmed by analyzing the parity of occupied Bloch wave functions at the time-reversal invariant points in Brillouin zone, which undoubtedly support gapless Dirac-type surface states inside the band gap (Fig. 1b)." It is debatable whether the authors verified the value of Z_2 independently. If not, it is recommended that this sentence be eliminated or a source referenced.

Authors' response

Thanks for the reviewer's professional suggestion. In fact, we indeed verified the value of Z_2 independently by calculating the parity of occupied Bloch wave functions at the time-reversal invariant points in Brillouin zone. Based on the detailed analysis, we found the parity of the occupied bands is negative at Γ point and positive for other points, thus resulting in a $Z_2 = 1$ topological insulator. Our calculated results are consistent with the previous report (*Phys. Rev. Lett.* 2011, 106, 156808). In our revision, we have added this source reference (*Phys. Rev. Lett.* 2011, 106, 156808) to the as-mentioned sentence.

Reviewer #3 (Remarks to the Author):

I found that the authors nicely revised the manuscript and included the new data requested by the referees. Although I think that the electrical resistivity in the conventional high mobility semimetals is very sensitive to the magnetic field direction when the angle is changed from the out-of plane to the current direction (anisotropy is comparable with the magnetoresistance ratio), I understand the mechanism in the present system is totally different and the results are novel.

I recommend the publication of the paper.

Authors' response

We highly appreciate the reviewer's positive evaluations on our efforts to revise our paper. Yes, just as pointed by the reviewer, the PHE and in-plane AMR show the totally different mechanism with the out-of-plane magnetoresistance. In our work, we reported the first observation of PHE and in-plane AMR in CVD-grown β -Ag₂Te, and fabricated the first topological-insulator based AMR sensors with a small device size and high performance. Thanks again for your constructive suggestions and inspiring questions raised during last-round revision, which really help us to improve the quality of our work.

REVIEWERS' COMMENTS

Reviewer #2 (Remarks to the Author):

The authors have improved the discussion concerning the impact of their findings, and I now recommend the publication of this manuscript in nature communication.